# Model-Based Underwater Image Simulation and Learning-Based Underwater Image Enhancement Method

**Yidan Liu \*, Huiping Xu, Bing Zhang, Kelin Sun, Jingchuan Yang, Bo Li, Chen Li and Xiangqian Quan**

Institute of Deep-Sea Science and Engineering, Chinese Academy of Sciences, Sanya 572000, China; xuhp@idsse.ac.cn (H.X.); zhangb@idsse.ac.cn (B.Z.); sunkl@idsse.ac.cn (K.S.); yangjc@idsse.ac.cn (J.Y.); libo@idsse.ac.cn (B.L.); lic@idsse.ac.cn (C.L.); quanxq@idsse.ac.cn (X.Q.)
* Correspondence: liuyd@idsse.ac.cn

**Abstract:** Due to the absorption and scattering effects of light in water bodies and the non-uniformity and insufficiency of artificial illumination, underwater images often present various degradation problems, impacting their utility in underwater applications. In this paper, we propose a model-based underwater image simulation and learning-based underwater image enhancement method for coping with various degradation problems in underwater images. We first derive a simplified model for describing various degradation problems in underwater images, then propose a model-based image simulation method that can generate images with a wide range of parameter values. The proposed image simulation method also comes with an image-selection part, which helps to prune the simulation dataset so that it can serve as a training set for learning to enhance the targeted underwater images. Afterwards, we propose a convolutional neural network based on the encoder-decoder backbone to learn to enhance various underwater images from the simulated images. Experiments on simulated and real underwater images with different degradation problems demonstrate the effectiveness of the proposed underwater image simulation and enhancement method, and reveal the advantages of the proposed method in comparison with many state-of-the-art methods.

**Keywords:** underwater image simulation; underwater image enhancement; modeling of underwater image degradation; encoder-decoder; deep learning

## 1. Introduction

Clear underwater images can provide first-hand information about the structure, texture, and color of underwater scenes and objects, and are vital for many underwater operations, such as marine archeology, seafloor mapping, and marine biological research [1–5]. However, due to the complexity of underwater environment, underwater images are prone to several degradation problems [6,7], which impact their utilities in underwater applications. Firstly, underwater images are affected by the wavelength-dependent attenuation of light in the water body, and present different degrees of color deviation depending on light propagation distance and water body property. Secondly, underwater images are degraded by the scattering effects caused by particles in the water body. These particles alter the directions of light, and cause blurriness and contrast reduction in underwater images. In shallow water, the refraction of light rays by the water surface may even cause caustics, which alters the actual appearance of objects of interest [8,9]. Moreover, illumination deficiency caused by light attenuation and non-uniform lighting is also a non-negligible issue, because in regions with insufficient lighting, the visibility of details is often limited. This issue is especially important for images in the deep sea, where non-uniform artificial illumination is the only light source. In Figure 1, we present four samples of underwater images with different degradation conditions for demonstration. Here, Figure 1a–c present degradation problems of bluish color deviation, contrast reduction, and insufficient illumination, respectively, and Figure 1d is an example of a clear underwater image with no

degradation. All these images are from the unpaired real-world underwater image subset of the EUVP dataset [10].

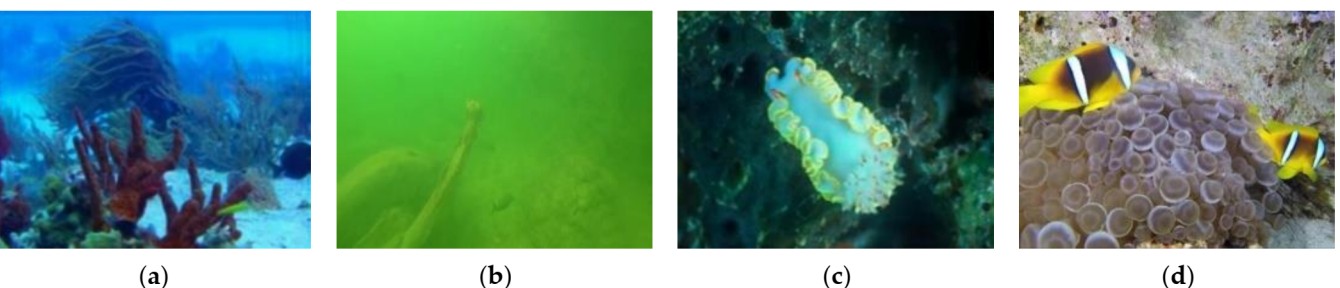

| (a) | (b) | (c) | (d) |

**Figure 1.** Underwater images with different degradation conditions. (**a**) An underwater image with bluish color deviation. (**b**) An underwater image with low contrast. (**c**) An underwater image with insufficient illumination. (**d**) An underwater image with no visible degradation.

Due to the complexity of underwater image degradation problems, the enhancement of underwater images has been a challenging task for a long time. Recently, owing to its great success in many vision tasks, deep learning was introduced in the field of underwater image enhancement. However, unlike many other tasks, the lack of paired training data is still a big obstacle for underwater image enhancement, because ground-truth images, i.e., clear counterparts of degraded underwater images, are almost impossible to get in the real world. There have been several attempts to solve this problem, such as generating simulated underwater images from clear images by using underwater image models [11] or deep-learning networks [10], or producing ground-truth candidates for real underwater images by using existing underwater image enhancement methods [12], but to the best of our knowledge, most of these methods are still limited to common degradation problems such as bluish color deviation and water-caused scattering, and have little control to the quality of generated image sets. As a result, the generated image sets can have narrower or different distributions than real-world underwater images, and methods using these sets as training data tend to have poor performance in solving diverse degradation problems in real-world underwater images.

In this paper, we propose a complete method of underwater image enhancement based on model-based image simulation and deep learning to deal with diverse degradation problems in real underwater scenes. To generate reliable training sets for deep learning, we derive a novel image model to describe the various degradation problems in real water and propose an underwater image simulation and selection method for generating reliable training sets for the targeted real-world images. Then, we propose a convolutional neural network (CNN) based on the encoder-decoder backbone with skip connections [13] and shortcut connections [14] to perform direct translations from degraded images to clear images. The proposed network is trained with the training sets generated by the aforementioned method to deal with various degradation problems in real-water underwater images. Experiments on underwater images with different degradation problems demonstrate the effectiveness and advantages of the proposed method, especially in enhancing images from challenging underwater scenes, such as the deep sea with insufficient lighting. The main contributions of this paper are summarized below:

- We propose a sophisticated underwater image simulation and selection method for generating training sets for learning-based underwater image enhancement methods. To the best of our knowledge, the proposed method covers a wider range of degradation problems than previous methods and is the first one that can simulate underwater images with uneven lighting and generate training sets specialized for the targeted real-world images.
- We propose an efficient CNN to learn the translation from degraded underwater images to clear enhanced images. Owing to the smooth transmission of information in

the proposed network, textures and structures of the original underwater images can be well-preserved in the enhanced images.

- The proposed method also outperforms many state-of-the-art methods, especially in enhancing underwater images with challenging degradation problems, such as strong light scattering or insufficient lighting.

## 2. Related Works

Underwater image degradation is a complex process that is influenced by many factors. However, most of them are hard to know during the underwater image enhancement process, which makes it an underdetermined problem that requires additional constraints to solve. Here, based on the types of these constraints, we classify underwater image enhancement methods into three categories and carry out a brief review of each of them.

### 2.1. Methods Based on General Image Processing Skills

The first type of underwater image enhancement method uses general image processing skills to improve the quality of underwater images. As mentioned before, the degradation of underwater images can appear in different aspects, including color, texture, contrast, and brightness. Methods of this type are built according to the aspects of image degradation. In 2007, Iqbal et al. [15] proposed an approach to improve the perception of underwater images. They performed contrast stretching on the RGB color space for color contrast equalization, and saturation and intensity stretching on HSI color space to increase the true color and address the lighting problem. Ancuti et al. [16] proposed a fusion-based method to solve the color deviation and contrast reduction problem. Their method combined several inputs generated by white balance, min-max windowing, and their combination, by weight maps of image luminance, contrast, chroma, and saliency. In Ancuti et al.'s later work [17], they proposed a new white balance method, and introduced gamma correction and unsharp masking to further improve the color balance, global contrast, and edge sharpness of input images. Fu et al. [18] used the Retinex theory to decompose the reflectance and illumination of the image, and applied histogram equalization and specification to address the fuzz and under-exposure problems. Similarly, Zhang et al. [19] also used the Retinex theory to obtain the reflectance and illumination component, but they used the simpler gamma correction for illumination adjustment.

### 2.2. Methods Based on Physical Models

In the second type of method, the complicated underwater image degradation process is described with simplified models and solved as an inverse problem by using statistics and inherent characters of underwater images as constraints. In these methods, the simplified image formation model (IFM) [20] is often used. As shown in Equation (1), IFM models underwater image $I^k(x)$ as the linear combination of clear image $J^k(x)$ and background light $B^k$ weighted by transmission map $t^k(x)$, which is calculated by $t^k(x) = e^{-c^k d(x)}$, where $c^k$ is attenuation coefficients and $d(x)$ is transmission distance.

$$I^k(x) = J^k(x)t^k(x) + B^k\left[1 - t^k(x)\right], \quad k \in \{R, G, B\},  \tag{1}$$

Due to the similarity of IFM and the hazy image model in the dark channel prior-based (DCP) haze-removal method in [21], the idea of DCP is also borrowed in underwater image enhancement. In [22], Galdran et al. proposed a red channel prior method to restore the colors associated to short wavelengths in underwater images. Drews Jr. et al. [23] proposed an underwater DCP that utilizes blue and green color channels as the underwater visual information source in the enhancement of underwater images. Carlevaris-Bianco et al. [24] proposed a prior based on the difference of attenuation between different color channels and used it to estimate the scene depth in underwater images. Apart from DCP, other priors and assumptions of the underwater images are also used in this type of method. In [20],

Peng and Cosman proposed to exploit image blurriness and light absorption to estimate the background light and transmission map of the IFM, and restore underwater images with these parameters. Liu et al. [25] extended the IFM by assuming locally invariant illumination to incorporate the case with non-uniform light sources, and use the new model to solve the underwater degradation problems caused by the water body and the illumination. Marques et al. [26] utilized DCP to enhance low-lighting underwater images. More specifically, they applied DCP on the inverted underwater image to improve the brightness and contrast of low-lighting regions and proposed a contrast-guided approach for calculating dark channels and transmission maps with patches of varying sizes.

### 2.3. Methods Based on Deep Learning

Despite the diversity of aforementioned methods, most of them do not perform well in challenging cases. Recently, deep learning was introduced in the enhancement of underwater images due to its success in many other vision tasks. Unlike former two types of methods, learning-based methods use CNNs to capture the latent relationships between degraded and clear images in the training set, and enhance the targeted underwater images with trained networks, so the performance of learning-based methods is highly influenced by the training set in use. Since the clear counterparts of degraded underwater images are usually hard to get in the real world, generating reliable training sets is also a challenging task and is crucial for these methods. In [11], Li et al. proposed a lightweight CNN for underwater image enhancement. For preparing training sets, they used IFM and attenuation coefficients of Jerlov water types to simulate degraded images from 10 different water types. The same training set was also used in [27], where Uplavikar et al. proposed an all-in-one network that adversarially learns the domain agnostic features to generate enhanced underwater images from degraded images of 10 different water types. In [28], Wang et al. proposed a generative adversarial network (GAN) based on an improved IFM to generate underwater images. These images are then used to train a U-Net for underwater image enhancement. Li et al. [29] proposed another GAN for generating underwater images. Their method simulates attenuation and backscattering caused by the water body, as well as vignetting effect of the camera. In [30], Li et al. proposed a network structure with medium transmission-guided multi-color space embedding for coping with the color casts and low contrast problems of underwater image. In [31], Fabbri et al. proposed to use the cycle-consistent adversarial network (CycleGAN) [32] to directly learn to generate degraded underwater images based on two separate groups of clear and degraded real-world images. The generated images are then used to train a U-Net for underwater image enhancement. In [33,34], the same dataset was used to train a multiscale dense GAN and a multilevel feature fusion-based conditional GAN, respectively, for underwater image enhancement. In [10], Islam et al. utilized the same CycleGAN-based method to build a large dataset called EUVP, and proposed a lightweight conditional GAN for fast underwater image enhancement (FUnIE-GAN). Li et al. [12] constructed a real-world underwater image benchmark (UIEB) by using traditional underwater image enhancement methods to generate clear reference images from degraded underwater images. They also proposed a gated fusion network called Water-Net to learn to enhance underwater images. In general, training sets used in learning-based underwater image enhancement methods are mainly built from models, networks, or real-world images. To the best of our knowledge, UIEB is one of the most realistic and diverse training sets for underwater image enhancement, but it only contains 890 image pairs, which limits its utility in training large networks.

### 3. Methodology of Underwater Image Simulation

To develop a widely applicable learning-based method for underwater image enhancement, we start from the foundation problem of generating reliable training sets, for which we derive a new model to incorporate a wider range of degradation problems and propose a novel method for generating specialized training sets for various real-world

underwater images. The generated training sets can then be used to train the CNN that learns to translate degraded underwater images to their clear counterparts. An example of such CNN is given in the next section.

### 3.1. Proposed Underwater Image Degradation Model

According to McGlamery's and Jaffe's works [35,36], monochromatic irradiance received by a camera underwater is composed of the direct component, the forward scattering component, and the backscattering component. Since the forward scattering component has high computational complexity but relatively low influence on pixel values and limited function range, we removed this component from our model for simplicity as done in many previous works [37–39].

The direct component mainly describes the energy loss of light underwater caused by attenuation. Its formula in [35,36] is given by

$$E_d(x,y|\lambda) = E_I(x,y|\lambda)M(x,y|\lambda)e^{-c(\lambda)d(x,y)}\cos^4\theta(x,y)\cdot\frac{T_l}{4f_n^2}\cdot\left[\frac{d(x,y)-F_l}{d(x,y)}\right]^2, \quad (2)$$

where $E_d(x,y|\lambda)$ represents the irradiance of the direct component and $E_I(x,y|\lambda)$ is the irradiance of incident light underwater. $M(x,y|\lambda)$ is scene reflectance, $c(\lambda)$ is volume attenuation coefficient and $d(x,y)$ is transmission distance. $\theta$, $T_l$, $f_n$, and $F_l$ are parameters of system geometry and camera lens and are usually hard to know in image simulation. To remove these parameters, we assumed an identical imaging system in air with no water-caused attenuation and derived the following formula.

$$J(x,y|\lambda) = E_0(\lambda)M(x,y|\lambda)\cos^4\theta(x,y)\cdot\frac{T_l}{4f_n^2}\cdot\left[\frac{d(x,y)-F_l}{d(x,y)}\right]^2. \quad (3)$$

Here, $J(x,y|\lambda)$ is the received irradiance with no water, and $E_0(\lambda)$ is the irradiance of incident light in air. By combining Equations (2) and (3), the calculation of the direct component is simplified as

$$E_d(x,y|\lambda) = \frac{E_I(x,y|\lambda)J(x,y|\lambda)e^{-c(\lambda)d(x,y)}}{E_0(\lambda)}. \quad (4)$$

The backscattering component mainly contributes to the hazy looks of underwater images. In [40], Zhao et al. provided a simple but efficient formula for calculating this component, which is

$$E_{bs}(x,y|\lambda) = E_{bs,\infty}(\lambda)\left[1 - e^{-c(\lambda)d(x,y)}\right], \quad (5)$$

where $E_{bs,\infty}(\lambda)$ is the background light. Its formula under homogeneous illumination is given by

$$E_{bs,\infty}(\lambda) = \frac{\kappa_l E_s(\lambda)}{c(\lambda)}\int_\Theta \beta(\phi)d\phi, \quad (6)$$

where $\kappa_l$ is a constant of the system's properties, $E_s(\lambda)$ is the incident irradiance, $\beta(\phi)$ is the volume-scattering function, and $\Theta$ represents all possible scattering angles. To incorporate inhomogeneous illumination under simpler expression, we use $E_s(x,y|\lambda)$ to replace $E_s(\lambda)$, and use constant $\kappa$ to replace the product of $\kappa_l$ and $\int_\Theta \beta(\phi)d\phi$, i.e.,

$$E_{bs}(x,y|\lambda) = \frac{\kappa E_s(x,y|\lambda)}{c(\lambda)}\left[1 - e^{-c(\lambda)d(x,y)}\right]. \quad (7)$$

Finally, to derive a concise image model from the irradiance model, we assume a delta response function of the camera sensor and use it to integrate the total received irradiance

(i.e., the combination of Equations (4) and (7)) of all wavelengths. The derived model is given in Equation (8). We use similar symbol expressions as in IFM for better comparison.

$$I^k(x,y) = L^k(x,y)J^k(x,y)e^{-c^k d(x,y)} + \frac{\kappa L_s^k(x,y)}{c^k}\left[1 - e^{-c^k d(x,y)}\right], \quad k \in \{R,G,B\}. \quad (8)$$

The most significant difference between IFM and our new model is the lighting parameters $L^k(x,y)$ and $L_s^k(x,y)$, which represent the incident light intensities on the photographed scene and the scattering medium, and correspond to $E_I(x,y|\lambda)/E_0(\lambda)$ in Equation (4) and $E_s(x,y|\lambda)$ in Equation (7), respectively. Apparently, with these parameters, the new model is able to simulate more complicated illumination conditions and analyze their influence on underwater image degradation.

### 3.2. Generating Reliable Training Sets by Underwater Image Simulation and Selection

With the proposed model, a simulated underwater image can be easily generated by assigning values to corresponding parameters. However, considering the diversity of underwater image degradation problems in the real world, it is very hard to choose the right parameters in the simulation process. Therefore, we propose to adopt another strategy that first expands the value range of model parameters as much as possible, then selects simulated images according to their quality and reliability.

#### 3.2.1. Underwater Image Simulation with Broadened Ranges of Parameter Values

In this subsection, we introduce our strategies for broadening the value ranges of model parameters in the simulation process. Since spatially varying illumination is not included in previous simulation works [2,19,20], we especially proposed a method to effectively simulate variant illumination conditions in underwater scenes.

(1)    Simulation of lighting parameters $L^k(x,y)$ and $L_s^k(x,y)$

In real-world conditions, illumination of underwater images can be extremely complicated. Due to the absorption and scattering effects in the water body, incident light is often degraded before reaching the object, and thus brings initial color deviations to the underwater images. Furthermore, in the deep sea, artificial illumination is the only light source, and its non-uniformity often results in uneven brightness and limited visibility in the underwater images. To simplify the simulation of underwater lighting parameters, we use the combination of the following three basic patterns to approximate real-world underwater illumination: (a) uniform pattern with constant intensities for simulating shallow-water illumination, (b) parallel pattern whose intensities degrade with depth, and (c) artificial pattern that provides spatially uneven illumination. Diagrams and real-world examples of these patterns are presented in Figure 2. For simplification, we also assume $L^k(x,y)$ and $L_s^k(x,y)$ are identical and use parameter $\kappa$ to adjust the relative intensity of backscattering light.

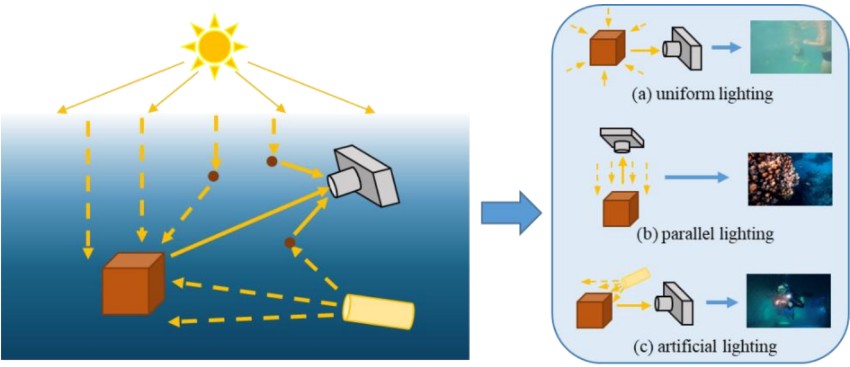

**Figure 2.** Approximating complicated real-world underwater illumination with the combination of uniform lighting, parallel lighting, and artificial lighting.

The proposed formula for simulating underwater illumination is shown in Equation (9). In the right half of Equation (9), the first part is the intensity of uniform lighting (i.e., pattern (a)), the second part is the intensity of parallel lighting (i.e., pattern (b)), and the last part is the intensity of artificial lighting (i.e., pattern (c)). Here we use $L_t^k$ and $\omega_t$ ($t \in \{a, b, c\}$) to represent the initial intensities and weights of these patterns, and their values are randomly selected from [0.9, 1.0] and [0, 1], respectively, in the simulation process. The sum of all weight parameters is equal to 1.

$$L^k(x,y) = \omega_a L_a^k + \omega_b L_b^k e^{-c^k(Z_b + d(x,y))} + \sum_i^{N_c} \omega_{c,i} \kappa_c \mathcal{P}\left(x, y \Big| L_{c,i}^k, \sigma_i\right) e^{-c^k D_{c,i}(x,y)} \qquad (9)$$

The distribution of parallel lighting follows the Beer–Lambert's Law. Therefore, its simulation requires the initial-light-intensity parameter, light-attenuation parameter, and light-traveling-distance parameter. We randomly picked a light-camera distance $Z_b$ from [0, 2 m] and added it to the transmission distance $d(x,y)$ to approximate the light-traveling-distance parameter. The simulation method of light-attenuation parameter $c^k$ will be covered in the following part. The simulation method of transmission distance $d(x,y)$ is given afterwards.

The simulation of artificial lighting is the most complicated part. In real-world underwater conditions, the distribution of artificial lighting can be influenced by the number, position, orientation, and beam pattern of light sources, which makes it difficult to simulate quickly. For simplification, we fixed the orientation of the light sources to orthographic projection, and used 2D Gaussian distribution to approximate real beam patterns. The difference of arbitrary and fixed light modes is depicted in Figure 3. Clearly, by using these simplifications, the simulation of imbalanced artificial lighting can be done in a much easier way.

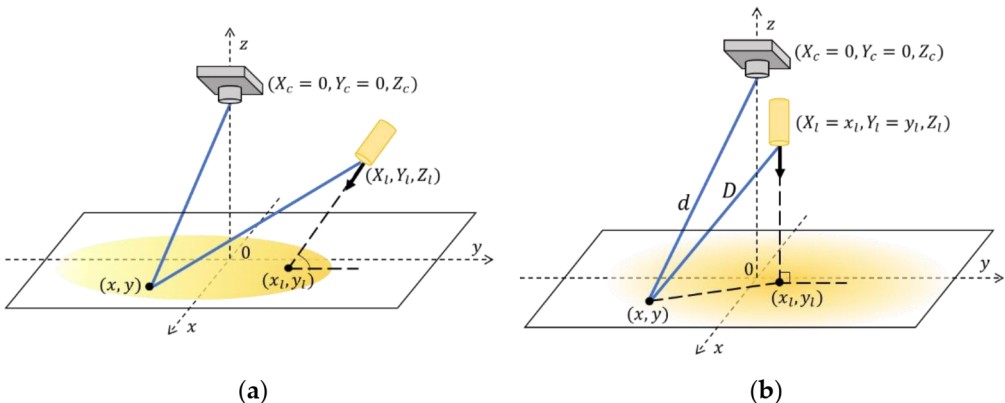

(**a**)                                        (**b**)

**Figure 3.** The difference of arbitrary and fixed light modes. (**a**) Diagram of artificial lighting with arbitrary orientation. (**b**) Diagram of artificial lighting with orthographic orientation. In the diagrams, $(X_l, Y_l, Z_l)$ represents the location of light source, $(x_l, y_l)$ represents the project of light center, $(X_c, Y_c, Z_c)$ is the location of camera, $(x, y)$ is the location of a photographed scene point, $d$ represents the distance from scene point to camera, $D$ is the distance from scene point to light source.

In the parameter setting of artificial lighting simulation, the total number of light sources $N_c$ is selected from [0, 2] according to the character of the targeted illumination pattern. The beam pattern $\mathcal{P}\left(x, y \Big| L_{c,i}^k, \sigma_i\right)$ is approximated by a Gaussian distribution with peak value at $L_{c,i}^k$ and standard deviation $\sigma_i$ proportional to the width of image by a random rate in [0.2, 1.1]. The distance from scene point to light source is calculated by $D_{c,i}(x,y) = \sqrt{Z_l^2 + r_l^2 \cdot \left[(x - x_{l,i})^2 + (y - y_{l,i})^2\right]}$, where $Z_l$ is the height of the $i$-th light source, and is randomly selected from [−1 m, 1 m], $(x_{l,i}, y_{l,i})$ is the projection of the $i$-th light source and is randomly picked from the pixel range, and $r_l$ is used to convert pixel distance to meter distance and is proportional to the ratio between the largest transmission

distance and the image width, the scaling parameter of which is randomly selected from [0.1, 1]. To ensure the validity of light intensity, we use a scalar $\kappa_c$ to adjust the intensity level, so that the highest value is equal to 1.

In Figure 4, five sets of simulated underwater lighting parameters are presented, which shows the ability of proposed method in simulating diverse illumination conditions, including uneven illumination with artificial light source in use.

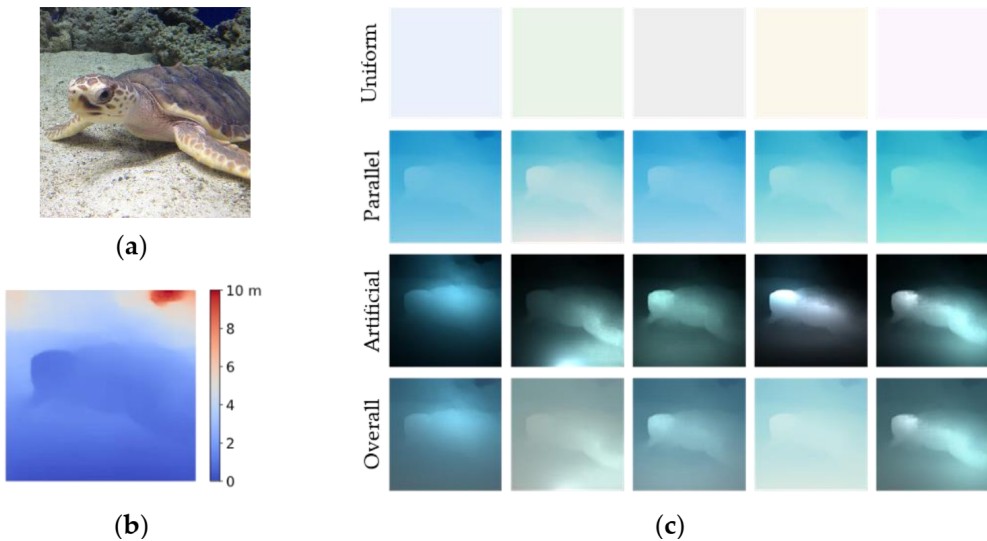

(a)

(b)　　　　　　　　　　　　　　　　　　　　(c)

**Figure 4.** Examples of simulated underwater lighting parameters. (**a**) A clear image used to calculate scene structure (taken from the EUVP dataset). (**b**) Simulated transmission distance $d(x, y)$ of the scene (method of simulating $d(x, y)$ is given in the following parts). (**c**) Simulation results of underwater lighting parameters generated by using the transmission distance $d(x, y)$ in (**b**).

(2)　Parameter setting of channel-wise attenuation coefficients $c^k$

To cover a wide range of water types in the simulation, we use attenuation coefficients of Jerlov water types [41] to generate values of channel-wise attenuation coefficients $c^k$, as in [11]. The Jerlov water types include two sets, where types I–III correspond to open ocean water and types 1–9 correspond to coastal water. From types I–III to 1–9, the turbidity of the water body grows gradually, and the color deviation of underwater images varies from blue to green to yellow. To generate $c^k$ from attenuation coefficient curves, a straightforward method is to pick one set of values at predefined response wavelengths as done in [40], but considering the diversity of camera sensors, we expand the predefined wavelengths to nearby intervals, and randomly select wavelength values in each simulation. Then, we linearly combine the attenuation coefficients of adjacent Jerlov water types and multiply it with random scalars, so as to introduce additional variation and simulate in-between water types of these Jerlov water types. Mathematically,

$$c^k = r_c^k \cdot \left[ \omega \cdot c_J\left(\lambda^k \middle| T_i\right) + (1 - \omega) \cdot c_J\left(\lambda^k \middle| T_{i+1}\right) \right] \tag{10}$$

where $c_J\left(\lambda^k \middle| T_i\right)$ represents the attenuation coefficient of Jerlov water type $T_i$ ($i \in [1, 7]$ corresponds to types I–III to 1–7, note that we do not include type IA and IB because they are very similar to type I.) at wavelength $\lambda^k$ (randomly selected within the $\pm 10$ nm intervals of response wavelengths in [40], i.e., $\lambda^R = 620$ nm, $\lambda^G = 540$ nm, and $\lambda^B = 450$ nm), $\omega$ is the random weighting parameter within [0, 1], and $r_c^k$ is the random scalar parameter from [0.9, 1.1]. Apparently, the proposed method can cover a wider range of attenuation coefficients than the Jerlov water type [41] itself.

(3)     Parameter setting of transmission distance $d(x, y)$

Instead of using RGB-D data to get paired $J^k(x, y)$ and $d(x, y)$ as in previous methods [11,28,29], we use megaDepth [42], a CNN model for single-view depth prediction, to estimate the initial transmission distance $d_0(x, y)$ from $J^k(x, y)$, and randomly scale it to get $d(x, y)$ for the simulation process, mathematically,

$$d(x, y) = r_{d1} + r_{d2} \cdot \left( \frac{d_0(x, y) - \min(d_0(x, y))}{\max(d_0(x, y)) - \min(d_0(x, y))} \right)^{r_{d3}}, \tag{11}$$

where $r_{d1}$, $r_{d2}$, and $r_{d3}$ are random scalars that are picked from [0, 5 m], [2 m, 10 m], and [0.5, 2], respectively. In this way, the diversity of scenes and distance ranges of the simulated images will not be limited by the finite RGB-D datasets, and thus increases the number and diversity of simulated underwater images.

(4)     Parameter setting of $\kappa$

As previously mentioned, $\kappa$ is a scalar about the camera system and backscattering. In the simulation, it is set as $\kappa = r_\kappa c^k / \max \left( L^k(x, y) \right)$ with $r_\kappa$ randomly selected from [0.3, 1], so as to prevent overexposure while controlling the proportion of backscattering.

(5)     Generating underwater image $I^k(x, y)$

After the simulation of all aforementioned parameters, underwater images can be generated by substituting corresponding parameters in Equation (8). In Figure 5, we present some samples of simulated underwater images generated by the proposed method. Obviously, the proposed method is able to generate various degradation appearances for a given image, including images with deviated color, low contrast, and insufficient lighting.

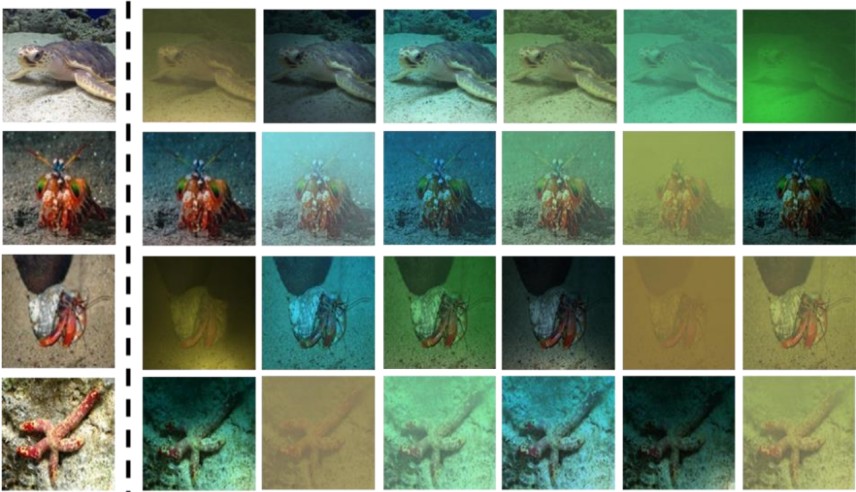

**Figure 5.** Samples of simulated underwater images generated by the proposed method. The first column presents clear images taken from the EUVP datasets, and the second to last columns are underwater images simulated based on the clear images in the first column.

Note that the diversity shown in Figure 5 is actually lower than the diversity the proposed method can reach, because the proposed method uses continuous values in most parameters. However, higher variance also indicates higher possibility to produce unrealistic simulated images, which should be excluded from the training process for higher efficiency. In the next subsection, we propose a simple but efficient method for selecting images into the training sets.

### 3.2.2. Generating Reliable Training Sets Based on Image Selection

We use the following two steps to select simulated underwater images into the training sets.

(1) Selecting simulated underwater images with similar color deviations as the targeted image sets

As shown in Figure 5, the variances of transmission distance, water type, and illumination condition have a big influence on the color deviation of a simulated underwater image. To ensure the similarity of the degradation patterns in the training image sets and the targeted image sets, we only select those simulated images with similar color deviations as the targeted images into the training sets. We use the frequency-based method in [25] to calculate the color tone of an image. In Table 1, we present the calculated color tones of four real-world underwater images and four simulated underwater images, together with their HSV values. Apparently, images with similar color deviations have similar H (hue) and V (brightness) values of their color tones. Therefore, by only selecting simulated images whose H and V values are in the intervals defined by the color tones of targeted images, we can reject simulated images with dissimilar color deviations to the targeted image set. The intervals of H and V values are defined by $[\sigma_{min}H_{min}, \sigma_{max}H_{max}]$ and $[\sigma_{min}V_{min}, \sigma_{max}V_{max}]$, respectively, where $H_{min}$ ($V_{min}$) and $H_{max}$ ($V_{max}$) are the minimum and maximum values of channel $H$ ($V$) of targeted images, $\sigma_{min}$ and $\sigma_{max}$ are scaling parameters, and are set according to the size of targeted image set. For small set, $\sigma_{min}$ and $\sigma_{max}$ are set as 0.5 and 1.5, respectively. Otherwise, their values are set to 1.

**Table 1.** Color tones and HSVs of real-world underwater images and simulated underwater images.

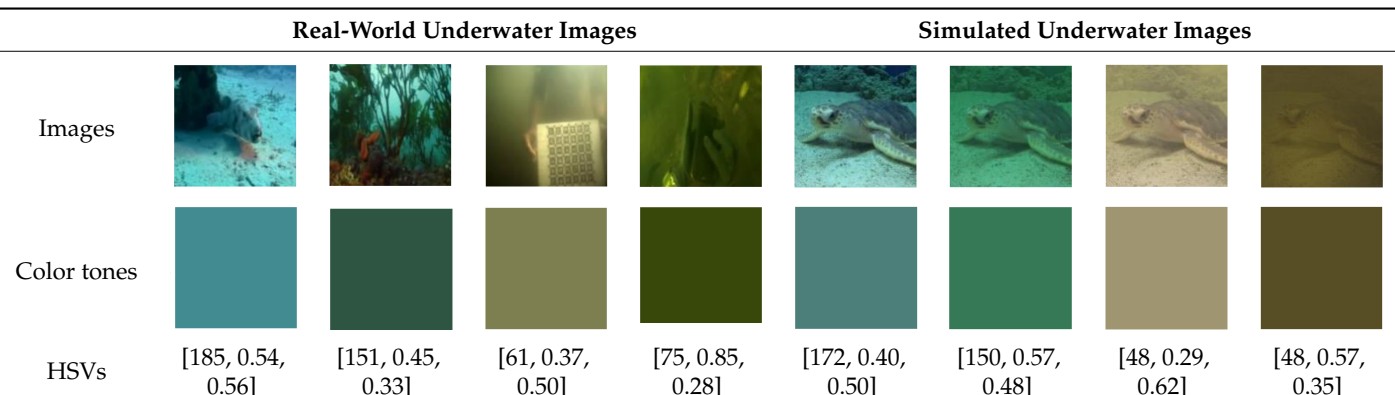

| | Real-World Underwater Images | | | | Simulated Underwater Images | | | |
|---|---|---|---|---|---|---|---|---|
| Images | | | | | | | | |
| Color tones | | | | | | | | |
| HSVs | [185, 0.54, 0.56] | [151, 0.45, 0.33] | [61, 0.37, 0.50] | [75, 0.85, 0.28] | [172, 0.40, 0.50] | [150, 0.57, 0.48] | [48, 0.29, 0.62] | [48, 0.57, 0.35] |

(2) Selecting simulated underwater images with enough details preserved

The preservation of enough details in the degraded underwater images is very important for providing necessary information in underwater image enhancement. To evaluate the degree of detail preservation in the degraded image, we use the Sobel edge map to quantify the intensity of details and calculate the difference between Sobel edge maps of the degraded image and the clear image. The Sobel edge map of an image is calculated as follows.

$$I_{edge}(x,y) = \sum_{k \in \{R,G,B\}} \left[ \begin{bmatrix} -1 & 0 & 1 \\ -2 & 0 & 2 \\ -1 & 0 & 1 \end{bmatrix} * I^k(x,y) \right]^2 + \sum_{k \in \{R,G,B\}} \left[ \begin{bmatrix} 1 & 2 & 1 \\ 0 & 0 & 0 \\ -1 & -2 & 1 \end{bmatrix} * I^k(x,y) \right]^2 \quad (12)$$

To avoid the influence of original detail intensities of the input image, we use the normalized mean absolute difference (NMAD) to evaluate the degradation degree of a simulated image, i.e., $\text{NMAD} = \sum_x \left| I_{edge}(x,y) - J_{edge}(x,y) \right| / \sum_x J_{edge}(x,y)$, where $J_{edge}$ is the edge map of the input clear image. Then, we exclude those simulated images with

NMADs higher than a predefined threshold from the training sets, so as to avoid artifacts caused by these images in the network training process. In practice, the threshold is set to 0.75. In Table 2, we present a clear input image and several simulated images with their Sobel edge maps and NMAD scores. Clearly, the proposed method mainly rejects simulated images with extremely low contrast or very limited visibility.

**Table 2.** Sobel edge maps of a clear image and simulated degraded images and their NMAD scores.

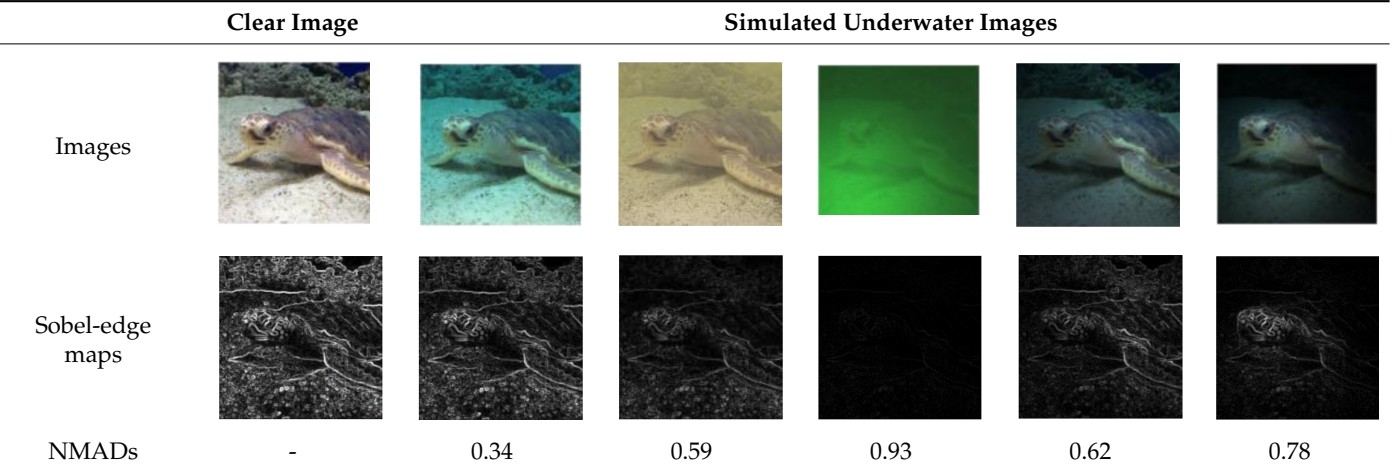

| | Clear Image | Simulated Underwater Images | | | | |
|---|---|---|---|---|---|---|
| Images | | | | | | |
| Sobel-edge maps | | | | | | |
| NMADs | - | 0.34 | 0.59 | 0.93 | 0.62 | 0.78 |

Apparently, the proposed selection method provides a simple way to generate specialized training sets according to the characters of targeted real-world underwater images. For underwater tasks within fixed regions, it helps to remove irrelevant degradation problems from the training sets, thus improves the efficiency of network training. In Table 3, we present some samples of three different training sets generated by the proposed method. In the simulation, the input clear images are from the non-degraded real-world images in the EUVP dataset [10], and the targeted real-world images are from the degraded real-world images in the EUVP dataset [10] and the UIEB dataset [12]. Here, the targeted images are specially selected from very different underwater scenes and grouped into three sets based on their degradation types, i.e., bluish color deviation in Set-A, strong scattering in Set-B, and insufficient lighting in Set-C. Obviously, the proposed method manages to simulate all the targeted degradation problems in the corresponding training set images, which verifies its effectiveness in generating specialized training sets for the targeted real-world images. To the best of our knowledge, it is the first method that can simulate underwater images with insufficient lighting and produce customized training sets for targeted underwater images.

**Table 3.** Sample images of three training sets generated by the proposed method.

| Group | Targeted Image Samples | Real-World Clear Images | Simulated Images after Selection |
|---|---|---|---|
| Set-A | | | |

**Table 3.** *Cont.*

| Group | Targeted Image Samples | Real-World Clear Images | Simulated Images after Selection |
|-------|------------------------|-------------------------|----------------------------------|
| Set-B | | | |
| Set-C | | | |

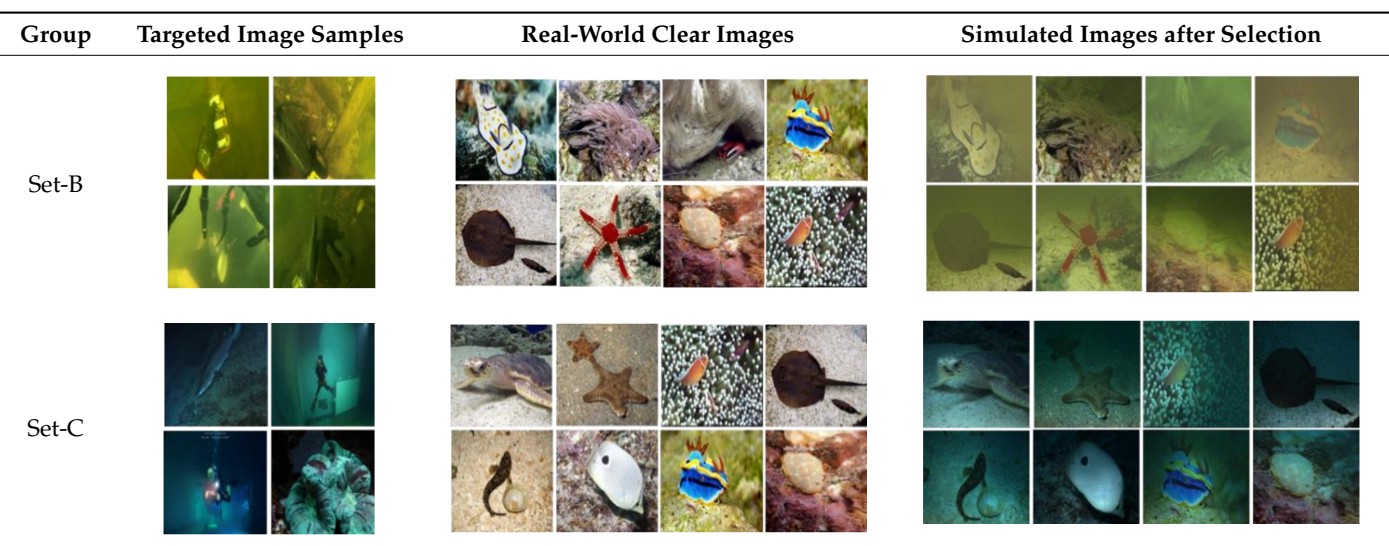

## 4. A Modular CNN for Learning to Enhance Degraded Underwater Images

In this section, we provide a detailed introduction of the architecture and object function of a modular CNN, which is to be trained with the generated training sets for enhancing targeted degraded images.

### 4.1. Network Architecture

The enhancement of a degraded underwater image requires to improve the overall quality of the image without losing useful information or introducing artifacts. To deal with the degradation problems presented in different image scales, we use the backbone of encoder-decoder structure to build a modular CNN to extract image features in different resolutions and construct the enhanced images. The architecture of the proposed network is shown in Figure 6.

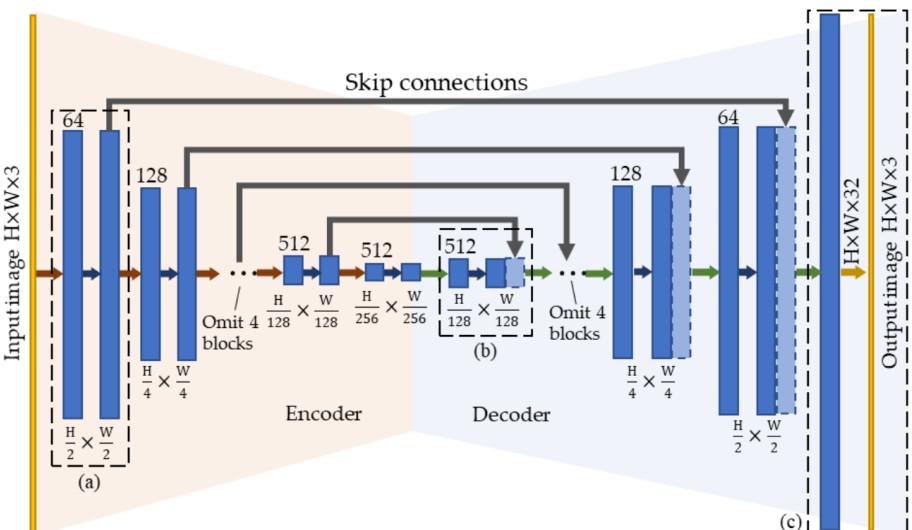

**Figure 6.** Architecture of the proposed underwater image enhancement network. Dotted-line frame (**a**) is an encoder block, (**b**) is a decoder block, (**c**) is the output block. All encoder blocks have same structure, and so are the decoder blocks but the last output block. Details of the encoder block, decoder block, and output block are given in Figure 7.

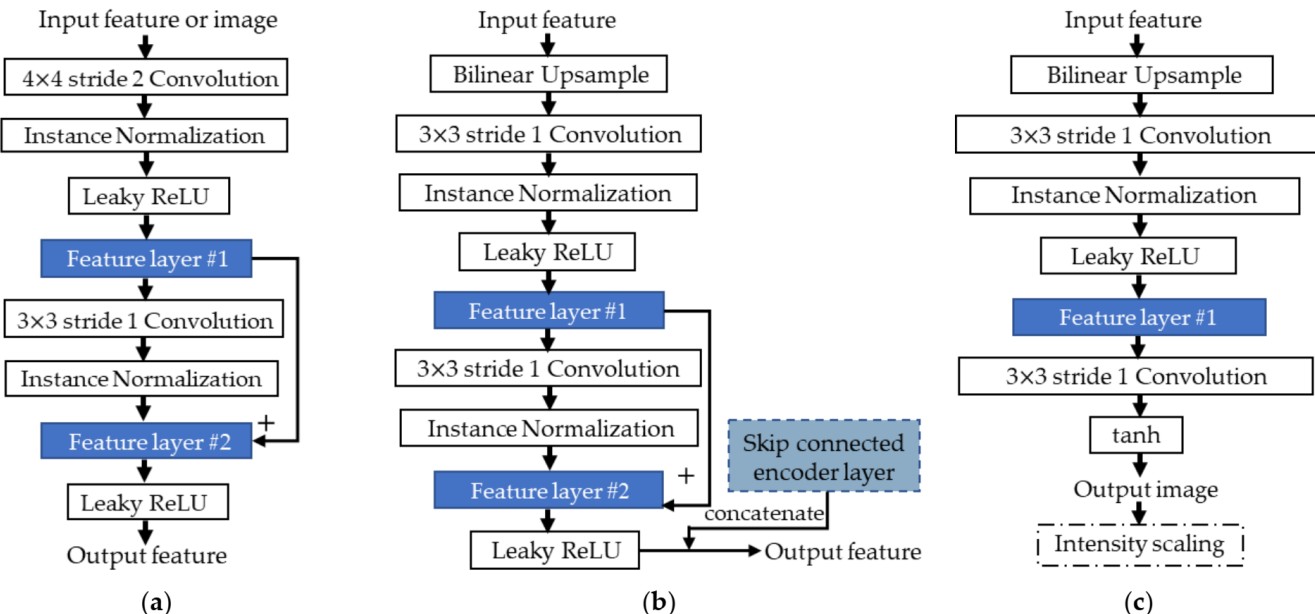

**Figure 7.** Detailed structure diagrams of basic blocks in the proposed network. (**a**) The structure of encoder block (the dotted-line frame (a) in Figure 6). (**b**) The structure of decoder block (the dotted-line frame (b) in Figure 6). (**c**) The structure of output block (the dotted-line frame (c) in Figure 6).

The proposed network consists of 8 encoder blocks and 8 decoder blocks, and each pair but the last one has a skip connection that sends the output of encoder block to decoder block. All encoder blocks have same structure, and so do the first 7 blocks in the decoder part. The last decoder block produces enhancement results and has a special structure. Details of the encoder block, common decoder block, and output decoder block are shown in Figure 7.

As shown in Figure 7, the encoder block contains two convolution layers together with two sets of instance normalization [43] and leaky ReLU [44] operations for feature map normalization and nonlinear activation. The negative slope of leaky ReLU is set as 0.2. The first convolution layer in the encoder block is $4 \times 4$ stride 2 convolution layer, which extracts feature maps with halved resolution from the input. The number of output feature maps of the first convolution layer starts from 64 and doubles gradually until reaching 512. After the first set of operations, we use a $3 \times 3$ stride 1 convolution layer for producing deeper features with same resolution and number. To ensure that useful information is well preserved in the block output, we also build an open path by shortcut connection [14] in this set of operations.

The decoder block is also composed of two sets of convolution, instance normalization, and leaky ReLU operations, but at the beginning of the first set, we add a bilinear interpolation operation to up-sample the input feature maps, then use a $3 \times 3$ stride 1 convolution to get deeper feature maps from the upscaled input. The number of output feature maps is changed inversely to that in the encoder part. Considering the possible information loss in the long path of feature map transmission in the backbone of the network, we also add skip connections to directly pass the output feature maps of the encoder blocks to the corresponding decoder blocks, as done in the U-Net [13]. More specifically, the skip connection creates a bypath for the feature maps produced by the encoder block so that they can be directly concatenated after the decoded feature maps of the same resolution as them and fed into the following decoder block of the next resolution level.

The output block is similar to the decoder block but uses tanh operation for nonlinear activation after the second convolution operation and directly generates output images

from it. The default value range of output images in the network is $[-1, 1]$, so an intensity scaling operation is added to change the value range to $[0, 1]$.

*4.2. Objective Function*

To improve the accuracy of the output images produced by the proposed network, we use a combination of a local loss function and a perceptually motivated loss function to constrain the network training process, which has been proven to be effective by experiments on similar vision tasks such as image denoising and demosaicking in [45].

The local loss function evaluates the pixel-wise difference between the network outputs and the ground-truth images, and mainly penalizes errors in color and brightness. In this work, we use $L_1$ loss function to calculate the local loss according to [45]. The formula of $L_1$ loss function is

$$L_1 = \frac{1}{N} \sum_x |I_{out}(x,y) - J(x,y)|, \tag{13}$$

where $I_{out}$ is the output image of the proposed network, $J$ is the ground truth image (i.e., the clear image in the simulation process), and $N$ is the total pixel number.

The perceptually motivated loss function is complementary to the local loss function and helps to improve the quality of output images in terms of characters sensitive to human visual systems. In this work, we use the structure similarity metric (*SSIM*) to evaluate the perceptual error of the image in the network training process. The *SSIM* of the output image with respect to the ground truth image is calculated as follows.

$$SSIM(x,y) = \frac{2\mu_{I_{out}}(x,y)\mu_J(x,y) + C_1}{\mu_{I_{out}}^2(x,y) + \mu_J^2(x,y) + C_1} \cdot \frac{2\sigma_{I_{out}J}(x,y) + C_2}{\sigma_{I_{out}}^2(x,y) + \sigma_J^2(x,y) + C_2} \tag{14}$$

Here, $\mu_{I_{out}}(x,y)$ and $\mu_J(x,y)$ are the average values of pixels intensities in a local region of $(x,y)$ in the output image and ground truth image, respectively. Similarly, $\sigma_{I_{out}}^2(x,y)$ and $\sigma_J^2(x,y)$ are the variances of these local region pixel intensities in the output image and ground truth image. $\sigma_{I_{out}J}(x,y)$ is the covariance of pixel intensities in these local regions of the output image and the ground truth image. The size of local regions is set as $11 \times 11$. $C_1$ and $C_2$ are constants and are set as $C_1 = (0.01L)^2$ and $C_2 = (0.03L)^2$, where $L$ represents the intensity range of images. Since the *SSIM* value grows with the improvement of image similarity, the loss function of *SSIM* is simply built as

$$L_{SSIM} = 1 - \frac{1}{N} \sum_x SSIM(x). \tag{15}$$

The final objective function is the linear combination of the $L_1$ loss and the *SSIM* loss, i.e.,

$$L = L_1 + L_{SSIM}. \tag{16}$$

## 5. Experimental Results

This section conducts a thorough evaluation of the proposed method, including the performance of proposed deep-learning network, the utility of proposed simulation method, and the performance of the whole enhancement method with both network and simulated data in use.

In the preparation of simulation dataset for network training, 10,350 clear images selected from EUVP dataset [10] and Microsoft COCO dataset [46] were used as inputs, and produced a simulation dataset with over 10,000 image pairs after selection. All images were regulated to the size of $256 \times 256$. The simulation dataset was split into training set and validation set by the ratio of 9:1. In the network-training process, the proposed network was trained over 200 epochs with the Adam optimizer ($\beta_1 = 0.9$, $\beta_2 = 0.999$). The batch size of training data was set as 24, and the learning rate was set as 0.0002. The proposed network was implemented on the PyTorch framework. All experiments were

conducted on a workstation with an Intel Xeon E5-1650 CPU, 48 GB RAM, and two NVIDIA TITAN X GPUs.

The performance of the proposed method was compared against three state-of-the-art learning-based methods and two non-learning methods, which are All-In-One [27], FUnIE-GAN [10], Water-Net [12], Liu's method [25], and Zhang's method [19]. The former three are learning-based methods, and are trained with Li's dataset [11], the EUVP dataset [10], and the UIEB dataset [12], respectively. The latter two are non-learning methods. More specifically, the second to last method is based on physical model, and the last method is built with general image processing skills. To control variables in the evaluation of proposed network and simulation method, networks from the three learning-based methods were retrained on the same simulation dataset produced by the proposed simulation method and generated three new network entities, named All-In-One*, FUnIE-GAN*, and Water-Net*, respectively.

### 5.1. Evaluation on Simulated Underwater Images with Various Degradation Problems

The first experiment was conducted on simulated underwater images produced by the proposed simulation method. To evaluate the performance of proposed network and simulation method thoroughly, all aforementioned learning-based methods and their retrained entities were included in this experiment, where the performance of proposed network was evaluated by comparing it with All-In-One*, FUnIE-GAN*, and Water-Net*, and the difference of the dataset produced by proposed simulation method and the three datasets used in training All-In-One, FUnIE-GAN, and Water-Net (i.e., Li's dataset, EUVP dataset, and UIEB dataset) was evaluated by comparing All-In-One, FUnIE-GAN and Water-Net with All-In-One*, FUnIE-GAN*, and Water-Net*, respectively. Underwater images used in this experiment cover a wide range of degradation problems, including color deviation, scattering, and insufficient lighting. In Figure 8, samples of input underwater images and the enhancement results are presented. For objective evaluation, the quality of enhance images were also evaluated by quantitative metrics, which were MSE (mean square error), PSNR (peak signal to noise ratio), and *SSIM*. The quantitative evaluation results are shown in Table 4.

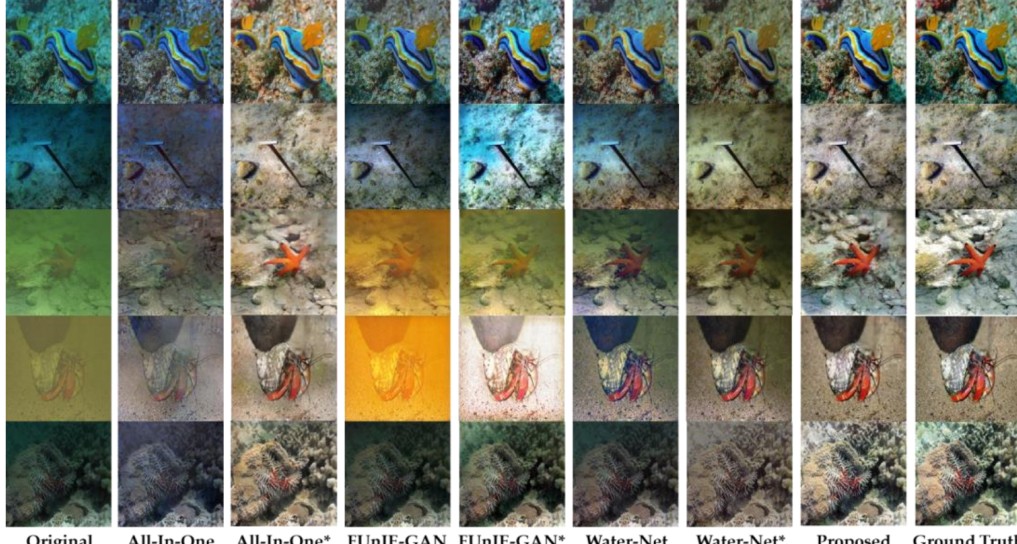

**Figure 8.** Qualitative evaluation on simulated underwater images. From left to right are original underwater images, enhanced images produced by each network entity, and the corresponding ground-truth images.

**Table 4.** Quantitative evaluation on simulated underwater images. ↓ means smaller scores are better. ↑ means larger scores are better.

|  | MSE↓ | PSNR↑ | *SSIM*↑ | Time (ms) |
|---|---|---|---|---|
| Original | 0.0217 | 16.6390 | 0.5343 | - |
| All-In-One | 0.0170 | 17.7063 | 0.6543 | 13.9641 |
| All-In-One* | 0.0053 | 22.7201 | 0.8050 | - |
| FUnIE-GAN | 0.0266 | 15.7483 | 0.5671 | **7.5361** |
| FUnIE-GAN* | 0.0211 | 16.7601 | 0.6969 | - |
| Water-Net | 0.0180 | 17.4526 | 0.6913 | 80.4657 |
| Water-Net* | 0.0176 | 17.5337 | 0.6894 | - |
| Proposed | **0.0029** | **25.4209** | **0.8616** | 27.7387 |

As shown in Figure 8 and Table 4, the proposed network produced images very similar to the ground truth and had much better scores than All-In-One*, FUnIE-GAN* and Water-Net*, which shared a same training set with the proposed network. Since the tested images were also produced by the proposed simulation method, this experimental result proves that the proposed network is able to learn the underlying relationship between the image pairs in the training set and has a better performance than the other three networks. The average runtime of each network is also presented in Table 4, which shows that the proposed network has a medium speed among all tested networks.

The difference between the simulation dataset produced by the proposed method and the three other datasets used in All-In-One, FUnIE-GAN and Water-Net is also very clear. As can been seen in Figure 8, images produced by entities with same network structure and different training data (i.e., All-In-One vs. All-In-One*, FUnIE-GAN vs. FUnIE-GAN*, Water-Net vs. Water-Net vs. Water-Net) are very different. In Table 4, entities trained with the simulation dataset produced by proposed method have better scores. This experimental result does not necessarily prove the superiority of the proposed simulation method, but reveals that the dataset produced by the proposed method has a different distribution with the other three dataset (i.e., Li's dataset [11], EUVP dataset [10], and UIEB dataset [12]).

### 5.2. Evaluation on Real-World Underwater Images with Various Degradation Problems

The second experiment evaluated the performance of proposed method on real underwater images. More specifically, the performance of the whole proposed method was evaluated by comparing it with all aforementioned methods in this section, including six network entities and two non-learning methods, and the applicability of different training datasets was assessed by comparing the performance of all learning-based methods. For reproducibility, all tested images in this experiment were collected from open sources such as published datasets (e.g., EUVP dataset [10] and the UIEB dataset [12]) and online sharing websites (e.g., Flickr https://www.flickr.com (accessed on 20 August 2020)). Samples of experimental results are shown in Figure 9. To evaluate the performance of each method in challenging cases, we especially select some severely degraded images in this experiment.

Since ground-truth images are not available, quantitative evaluation is conducted by using metrics that describe the color deviation, edge intensity, and information content of images, so as to quantify the improvement of image quality for images with different degradation problems after enhancement. The evaluation results of all tested methods are presented in Table 5. We did not adopt existing non-reference quality measurements of underwater, such as UCIQE [47] or UIQM [48], because their utilities are still in doubt, especially in evaluating underwater images with insufficient lighting [2,3]. The color deviation metric used in this experiment is from Li's work [49], where the average values and standart deviations of channel a and b of the Lab color space were used to evaluate the degree of color cast of an image. The edge intensity metric evaluates the richness of details in an image, and was obtained by calculating the average intensity of Sobel edge map of the image. The information content metric was evaluated by entropy, which evaluates the

richness of grayscales in the image. The latter two metrics are complementary in evaluating the visibility of details in the image.

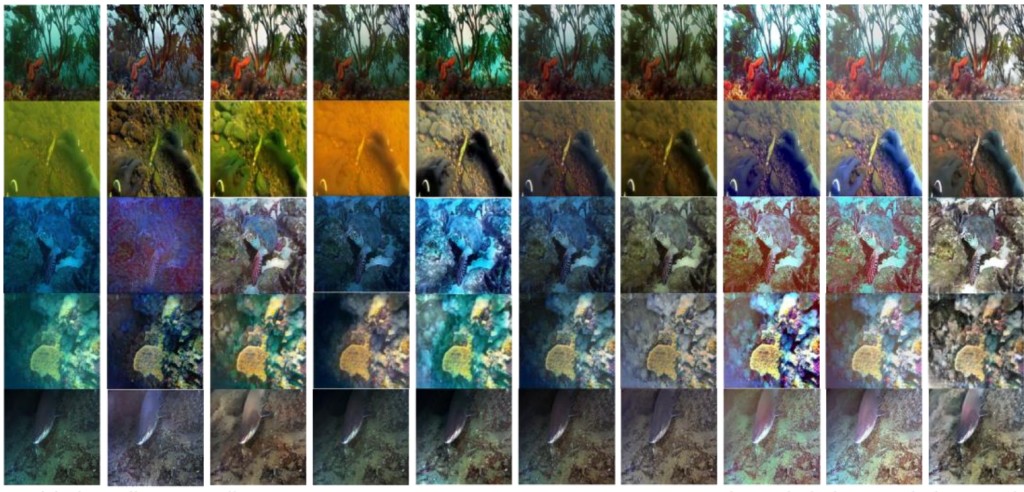

Original   All-In-One   All-In-One*   FUnIE-GAN   FUnIE-GAN*   Water-Net   Water-Net*   Liu's method   Zhang's method   Proposed

**Figure 9.** Qualitative comparisons on real-world underwater images. From left to right are original underwater images and enhanced images produced by each tested enhancement method.

**Table 5.** Quantitative evaluation on real-world underwater images. ↓ means smaller scores are better. ↑ means larger scores are better.

|  | Color Deviation↓ | Edge Intensity↑ | Information Content↑ |
|---|---|---|---|
| Original | 2.9042 | 0.1286 | 6.8242 |
| All-In-One | 1.5917 | 0.1941 | 6.9834 |
| All-In-One* | 1.2218 | 0.1965 | 7.2118 |
| FUnIE-GAN | 2.4229 | 0.1502 | 7.1567 |
| FUnIE-GAN* | 1.3253 | 0.2380 | 7.3171 |
| Water-Net | 0.8520 | 0.1603 | 7.2225 |
| Water-Net* | 1.0476 | 0.1701 | 7.0262 |
| Liu's method | 0.4312 | 0.2593 | 7.3229 |
| Zhang's method | **0.1132** | 0.2456 | 7.4529 |
| Proposed | 0.2988 | **0.2664** | **7.5247** |

In Figure 10 and Table 6, we also present a simple comparison of the UIQM metrics (including UICM for colorfulness, UISM for sharpness, UIConM for contrast, and UIQM for the whole image) and the three metrics used in this study. Sample images shown in Figure 10 are taken from Figure 5, including a high-quality image and three simulated degraded images. Clearly, the three metrics used in this study correctly assigned high scores to high-quality images, while the UISM metric seemed to prefer images with uneven lighting even though it does not have richer details.

As shown by Figure 9 and Table 5, the proposed method significantly improved the quality of input underwater images. Qualitatively speaking, it restored the deviated color tones in tested images, removed hazy looks, and improved the degrees of brightness and visibility in low-light regions of the tested images. The quantitative result in Table 5 is consistent with the qualitative result, where output images of the proposed method get much better scores than the original images.

The two non-learning methods, i.e., Liu's method [25] and Zhang's method [19], also improved the quality of tested images, but they tended to produce pseudo color in the enhancement results of images with strong color deviation, such as those in the 3rd and 4th rows, and could not fully restore the illumination balance in low-light images as shown in the 4th and last row of Figure 9. Accordingly, their scores of edge intensity and information content were lower than the proposed method. The color deviation score of

Liu's method [25] was also worse than the proposed method, and Zhang's method [19] obtained a better score in color deviation than the proposed method mainly due to the high color-richness of its result images.

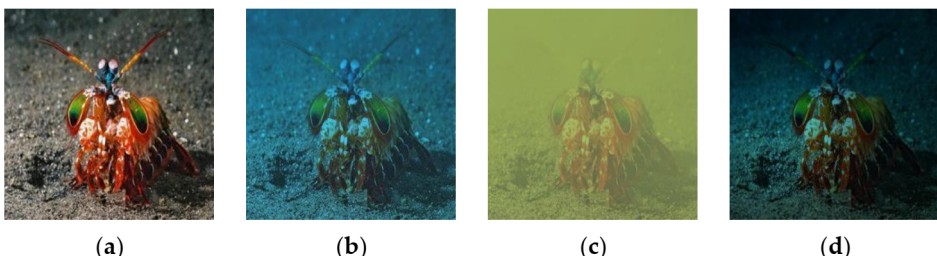

(**a**) (**b**) (**c**) (**d**)

**Figure 10.** Sample images used for comparing the UIQM metrics and the three metrics used in this study. These images are taken from Figure 5, where (**a**) is the high-quality image and (**b–d**) are simulated underwater images with various degradation problems. Corresponding evaluation results are presented in Table 6.

**Table 6.** Comparison of UIQM metrics and the three metrics used in this study on images from Figure 10. ↓ means smaller scores are better. ↑ means larger scores are better.

| Image | UICM↑ | UISM↑ | UIConM↑ | UIQM↑ | Edge Intensity↑ | Color Deviation↓ | Information Content↑ |
|---|---|---|---|---|---|---|---|
| Figure 10a | **6.1252** | 13.8233 | **0.2833** | 5.2677 | **0.1787** | **0.8622** | **7.2844** |
| Figure 10b | 3.6543 | 14.1887 | 0.2754 | **5.2775** | 0.1539 | 2.3381 | 7.1077 |
| Figure 10c | 0.8007 | 13.0062 | 0.2013 | 4.5832 | 0.0693 | 6.0097 | 6.2114 |
| Figure 10d | 3.6081 | **14.8173** | 0.1776 | 5.1124 | 0.1150 | 1.2231 | 6.4453 |

The three original learning-based methods, i.e., All-In-One [27], FUnIE-GAN [10] and Water-Net [12], did not perform well in this experiment. All-In-One and FUnIE-GAN failed in almost all tests and produced false color and textures in the enhanced images. Water-Net could not improve the visibility of low-light regions in the tested images. Their scores were also worse than those of the proposed method and the two non-learning methods.

The three retrained networks, i.e., All-In-One*, FUnIE-GAN*, and Water-Net*, performed better than the corresponding original networks but worse than the proposed method. In Figure 9, all result images of the retrained networks have more balanced color than those of their original counterparts, and All-In-One* and Water-Net* also managed to improve the visibility of low light regions in the tested images. The quantitative results of All-In-One* and FUnIE-GAN* are also better than their corresponding original network as shown in Table 5. Water-Net* sees better result in terms of edge intensity but worse scores in color deviation and information content, which might be due to its low color richness and small intensity range.

Overall, the proposed method shows its advantages against the competing methods in this experiment and is especially good at improving the visual quality of low-light underwater images. Networks trained with a simulation dataset generated by the proposed method also performed better, which implies the superiority of the proposed image simulation method in generating realistic images for training underwater image enhancement networks.

## 6. Conclusions

In this paper, we proposed a complete method for enhancing underwater images with different degradation problems. To utilize the superpower of deep learning, we first proposed a training set generating method to produce specified and reliable training data for the targeted underwater images, then proposed a convolutional neural network for translating degraded underwater images to their clear counterparts by learning from

the image pairs in the training set. Experiments on underwater images with different degradation problems demonstrated the effectiveness of the proposed method and its advantages against many state-of-the-art methods, especially in enhancing images from challenging underwater scenes, such as the deep sea with insufficient lighting. In the future, we plan to work on the improvement of the network structure and increase the diversity of image pairs in the training set, so as to further improve the efficiency and performance of the proposed method.

**Author Contributions:** Conceptualization, Y.L., H.X. and K.S.; methodology, Y.L.; software, Y.L. and C.L.; validation, B.Z., J.Y. and B.L.; formal analysis, X.Q.; investigation, B.Z. and B.L.; resources, H.X. and K.S.; data curation, Y.L. and C.L.; writing—original draft preparation, Y.L.; writing—review and editing, H.X. and K.S.; visualization, B.Z.; supervision, H.X.; project administration, C.L.; funding acquisition, B.Z. and K.S. All authors have read and agreed to the published version of the manuscript.

**Funding:** This research was funded by the National Key Research and Development Program of China, grant number 2018YFC0307905 and the National Natural Science Foundation of China, grant number 61972240.

**Institutional Review Board Statement:** Not applicable.

**Informed Consent Statement:** Not applicable.

**Data Availability Statement:** Not applicable.

**Acknowledgments:** The authors thank the associate editor and the anonymous reviewers for their constructive comments and suggestions.

**Conflicts of Interest:** The authors declare no conflict of interest.

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
