# Peer review of "Model-Based Underwater Image Simulation and Learning-Based Underwater Image Enhancement Method"

_information, doi:10.3390/info13040187_

Round 1
Reviewer 1 Report
it is a very interesting procedure to enhance the quality of underwater images.
The authors shown a good knowledge of the state of the art and they suggested an approach that is an enhancement of actual methods.
I suggest for the future to test the effect on image geometry of CNN model . By this test it should be used the procedure ven for high accuracy photogrammetric surveying.
Author Response
Thank you for your comments and suggestions, they are very useful for our current and future research. The modeling and rectifying of underwater image deformation is a quite challenging problem, because the PSF of underwater light rays can be influenced by not only ordinary forward scattering effect, but also factors like fine dust and hear flow that can be quite common in certain regions. We believe deep learning is a promising method in solving the problem of underwater image deformation, but we are still struggling in collecting paired images for network training. We are also planning to produce paired training data by simulation, but we have not found proper models yet. As long as the problem of training data is solved, the study of underwater photogrammetry would be feasible for us.
Reviewer 2 Report
Although the paper is rather specialistic, I found it interesting in reading. although some parts, especially regarding the selection of parameters for generating simulated images can be shortened and, at the same time, made more effective.
I have the following suggestions:
- The first lines of the introduction (1-44) lack proper bibliographic references.
- -In the introduction, among the different problems of underwater images caustics seem to have been overlooked. I understand the authors are more focused on deep water, however, such a problem should be mentioned also in relation to existing simulation methods to generate caustics and existing deep learning algorithms for their removal. It is a proper part of image enhancement. I suggest this paper https://scholar.google.com/scholar_lookup?title=DeepCaustics:+Classification+and+Removal+of+Caustics+From+Underwater+Imagery&author=Forbes,+T.&author=Goldsmith,+M.&author=Mudur,+S.&author=Poullis,+C.&publication_year=2019&journal=IEEE+J.+Ocean.+Eng.&volume=44&pages=728%E2%80%93738&doi=10.1109/JOE.2018.2838939 and this review https://www.mdpi.com/2072-4292/13/1/22?type=check_update&version=3 from a mdpi journal.
- At line 387 you used sobel essentially as a measure of the level of details of the generated images. Perhaps, can you comapre to existing metrics for image quality with no reference both in the UW and general scenario? This is mandatory.
- Subsection 3.3 should be moved to proper full section (e.g. Secion 4).
Round 2
Reviewer 2 Report
The paper has been reworked as requested. IMHO the paper is now suitable for publication.